# Characterization of dengue virus 3'UTR RNA binding proteins in mosquitoes reveals that AeStaufen reduces subgenomic flaviviral RNA in saliva

**Shih-Chia Yeh[1¤a☺], Mayra Diosa-Toro[1¤b☺], Wei-Lian Tan[1], Florian Rachenne[2], Arthur Hain[1], Celestia Pei Xuan Yeo[1], Inès Bribes[2], Benjamin Wong Wei Xiang[1], Gayathiri Sathiamoorthy Kannan[3], Menchie Casayuran Manuel[1], Dorothée Missé[2], Yu Keung Mok[3], Julien Pompon[1,2¤c]** *

**1** Programme in Emerging Infectious Diseases, Duke-NUS Medical School, Singapore, Republic of Singapore, **2** MIVEGEC, Univ. Montpellier, IRD, CNRS, Montpellier, France, **3** Department of Biological Sciences, National University of Singapore, Singapore, Republic of Singapore

¤a Current address: Cancer Science Institute of Singapore, National University of Singapore, Singapore, Republic of Singapore
¤b Current address: Department of Biomolecular Health Sciences, Faculty of Veterinary Medicine, Utrecht University, Utrecht, The Netherlands
¤c Current address: MIVEGEC, Univ. Montpellier, IRD, CNRS, Univ. Montpellier, Montpellier, France
☺ These authors contributed equally to this work.
* Julien.pompon@ird.fr

**Data Availability Statement:** All relevant data are within the manuscript and its Supporting Information files.

## Abstract

Dengue viruses (DENV) are expanding global pathogens that are transmitted through the bite of mosquitoes, mostly *Aedes aegypti*. As RNA viruses, DENV rely on RNA-binding proteins (RBPs) to complete their life cycle. Alternatively, RBPs can act as restriction factors that prevent DENV multiplication. While the importance of RBPs is well-supported in humans, there is a dearth of information about their influence on DENV transmission by mosquitoes. Such knowledge could be harnessed to design novel, effective interventions against DENV. Here, we successfully adapted RNA-affinity chromatography coupled with mass spectrometry–a technique initially developed in mammalian cells–to identify RBPs in *Ae. aegypti* cells. We identified fourteen RBPs interacting with DENV serotype 2 3'UTR, which is involved in the viral multiplication and produces subgenomic flaviviral RNA (sfRNA). We validated the RNA affinity results for two RBPs by confirming that AePur binds the 3'UTR, whereas AeStaufen interacts with both 3'UTR and sfRNA. Using *in vivo* functional evaluation, we determined that RBPs like AeRan, AeExoRNase, and AeRNase have pro-viral functions, whereas AeGTPase, AeAtu, and AePur have anti-viral functions in mosquitoes. Furthermore, we showed that human and mosquito Pur homologs have a shared affinity to DENV2 RNA, although the anti-viral effect is specific to the mosquito protein. Importantly, we revealed that AeStaufen mediates a reduction of gRNA and sfRNA copies in several mosquito tissues, including the salivary glands and that AeStaufen-mediated sfRNA reduction diminishes the concentration of transmission-enhancing sfRNA in saliva, thereby revealing AeStaufen's role in DENV transmission. By characterizing the first RBPs

**Funding:** Funding for this research came from a National Medical Research Council ZRRF grant (ZRRF/007/2017) and a French Agence Nationale de la Recherche grant (ANR-20-CE15-0006) both awarded to JP, a Rubicon scholarship from the Dutch Research Council (NOW) awarded to MD, a PhD fellowship from the French Ministry of higher education and research awarded to FR and the Emerging Infectious Diseases (EID) Signature Research Programme at Duke-NUS funded by the Agency for Science Technology and Research (A*STAR). The funders had no role in study design, data collection and analysis, decision to publish or preparation of the manuscript.

**Competing interests:** The authors have declared that no competing interests exist.

that associate with DENV2 3'UTR in mosquitoes, our study unravels new pro- and anti-viral targets for the design of novel therapeutic interventions as well as provides foundation for studying the role of RBPs in virus-vector interactions.

## Author summary

Dengue viruses are important human pathogens transmitted by mosquitoes. Currently, there are no effective control measures for dengue. The RNA-binding proteins (RBPs) in mosquitoes, which bind to the dengue virus genome to regulate viral multiplication, could serve as new targets for developing therapeutic interventions. In this study, we pioneered the use of RNA-affinity chromatography–a technique that identifies proteins binding to specific RNA sequences–in mosquito cells. This led to the detection of fourteen RBPs that associate with the 3'UTR of dengue virus serotype 2. We validated our results using immunoprecipitation method. Furthermore, we demonstrated that 6 of the 14 RBPs influence viral multiplication in mosquitoes. Among these six RBPs, we showed that the AePur mosquito and human homologs share an affinity to dengue virus RNA, whereas the antiviral function is specific to the mosquito homolog. Importantly, we revealed that AeStaufen mediates a reduction of genomic and subgenomic flaviviral RNAs in multiple mosquito tissues. We also showed that the reduction of subgenomic flaviviral RNA in salivary glands diminishes the secretion of salivary subgenomic RNA, which facilitates infection at the bite site, thereby unveiling the function of AeStaufen in the virus transmission. By characterizing the first mosquito RBPs that interact with dengue virus genome, our study paves the way for leveraging these proteins as potential targets to block virus transmission.

## Introduction

Dengue viruses (DENV) are transmitted to humans through mosquito bites, primarily of the *Aedes aegypti* species [1,2]. Given the wide geographic distribution of the mosquito vector and its continuous expansion, almost half of the human population is at risk of infection and about 400 million infections occur every year [3,4]. There are no approved therapeutics for dengue and the only licensed vaccine (DENGVAXIA) has highly variable efficacy against the four DENV serotypes. Moreover, it does not protect against primary infection, considerably limiting its uptake by the population [5]. Broadly-deployed vector control measures based on source reduction and insecticide treatments do not sustainably reduce dengue incidence, even upon sustained and thorough implementation of WHO recommendations [6]. An improved understanding of the molecular interactions that mediate successful viral transmission by mosquitoes is necessary to unravel new targets for the design of effective interventions.

DENV possess a positive-sense single-stranded RNA genome (gRNA) that serves multiple purposes in the viral life cycle. DENV gRNA is translated into viral proteins, functions as a template for RNA replication via the synthesis of complementary negative strands and is assembled into new virions. In all these processes, the gRNA interacts with host RNA-binding proteins (RBPs), which can act as pro or anti-viral factors [7–10]. In humans, several pro-viral RBPs that bind throughout the DENV genome have been identified [11,12]. Furthermore, there is considerable interest in RBPs that interact with the 3'UTR, as the non-coding region is key for viral replication, translation, and assembly [7]. Indeed, DENV replication requires the 3'UTR to interact with pro-viral factors such as polypyrimidine tract-binding protein (PTB),

NF90, NFκB2, LSm-1, Dead-box helicase 6 (DDX6), and Exoribonuclease family member 3 (ERI3) [13–18], while its translation and assembly are dependent on 3'UTR binding to two other pro-viral RBPs, poly-A-binding protein (PABP) [19] and YBX1 [20], respectively. Alternatively, the binding of DENV 3'UTR to Quaking, an anti-viral factor, restricts viral multiplication [21]. All these studies, conducted in human cells, demonstrate the importance of RBPs in DENV cellular cycle. However, there is no information on RBP interaction with the DENV genome or the 3'UTR in mosquitoes.

As in all other flaviviruses, the DENV gRNA is partially degraded by 5'-3' exoribonucleases such as Xrn1 that get stalled at nuclease-resistant structures present in the 3'UTR [22,23]. The abortion of the RNA decay process leaves a highly structured RNA fragment, corresponding to a partial 3'UTR sequence, referred to as subgenomic flaviviral RNA (sfRNA). DENV sfRNA is known to function as an immuno-suppressor via its interaction with several RBPs [24]. In humans, DENV sfRNA interacts with TRIM25 to inhibit signaling of the anti-viral interferon pathway [25] and with G3BP1, G3BP2, and CAPRIN1 to downregulate the translation of interferon-stimulated genes (ISG) [26]. In mosquitoes, DENV sfRNA inhibits the expression of components of the Toll immune pathway to promote viral transmission [27]. Furthermore, we recently revealed a new function for sfRNA at the interface between mosquitoes and humans. We showed that sfRNA is secreted in mosquito salivary vesicles to enhance saliva-mediated infectivity in human skin cells and promote infection at the bite site [28]. The multiple functions of sfRNA in viral transmission has accentuated interest in identifying RBPs that bind to DENV 3'UTR in mosquitoes.

In this study, we aim to identify RBPs that bind to DENV 3'UTR in mosquitoes and to characterize their functions in DENV transmission. Using RNA-affinity chromatography coupled with quantitative mass spectrometry (MS), which was adapted from earlier studies on RBP-3'UTR interactions in human cells [18,21,25], we identified fourteen proteins that interact with the 3'UTR of DENV serotype 2 (DENV2) in *Ae. aegypti* cells. We validated these findings in infection condition by confirming that AePurine-rich element binding protein (AePur) associates with the genomic 3'UTR, whereas AeStaufen interacts with both 3'UTR and sfRNA. We then documented the effects of the fourteen proteins on *in vivo* mosquito infection, which revealed pro-viral functions for three RBPs (AeRan and two nucleases) and anti-viral functions for two other proteins (GTPase and AePur). We found that while AePur 3'UTR interaction is conserved in the human homologs, the anti-viral function is specific to the mosquito protein. Finally, we discovered that AeStaufen reduces both gRNA and sfRNA levels in multiple mosquito tissues, including salivary glands, and that AeStaufen-mediated RNA decay alters the amount of transmission-enhancing sfRNA in the saliva.

## Results

### Fourteen *Ae. aegypti* proteins interact with DENV2 3'UTR

To identify proteins interacting with DENV2 3'UTR, we adapted our previously-described approach of RNA-affinity chromatography coupled with mass spectrometry (MS) using mosquito cell lysates (Fig 1A). We generated a construct containing the T7 promotor, a tobramycin adapter sequence, and DENV2 3'UTR from the NGC strain. For control, we replaced the 3'UTR sequence with a size-matched NS2A gene sequence from DENV2, as previously described [18]. The constructs were transcribed *in vitro* and the RNA sequences were allowed to form secondary structures prior to being incubated with tobramycin-conjugated sepharose beads. The RNA-tobramycin bead conjugates were then incubated with lysates from uninfected Aag2 cell line, an *Ae. aegypti* cell line commonly used for studying anti-viral responses [29]. Eventually, we eluted RNA-bound proteins by adding an excess of tobramycin and

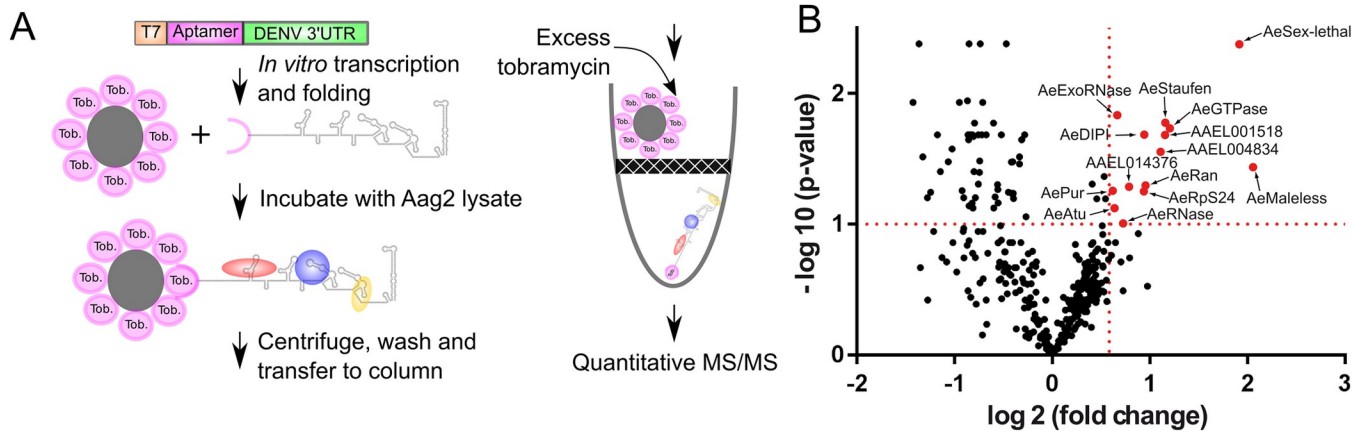

**Fig 1. Application of RNA-affinity chromatography to identify *Aedes aegypti* proteins that interact with DENV2 3'UTR.** (A) Schematic of the RNA-affinity chromatography coupled with quantitative mass spectrometry (MS). (B) Identification of the DENV2 3'UTR-bound proteins. Of the 385 proteins detected, 14 interacted more than 1.5 fold (p<0.1) with DENV2 3'UTR, as compared to the control RNA. Interacting proteins are shown in red and their names are indicated with an arrow.

quantified them using quantitative MS. Of note, we chose not to use infected Aag2 lysates to prevent competition between the virus-produced 3'UTR and sfRNA and the "artificial" 3'UTR introduced for RNA-affinity. Nonetheless, to confirm the RBP-viral RNA interactions identified by RNA-affinity chromatography, we subsequently used RNA immunoprecipitation (RIP) in infected mosquitoes and cells, as detailed in the following subsection.

Among the 385 proteins detected in the eluates (S1 Table), we defined DENV2 3'UTR-interacting proteins as those enriched more than 1.5-fold over the NS2A RNA control, with an adjusted p-value < 0.1. We have identified 14 proteins that met these criteria (Fig 1B and Table 1), which include (in decreasing order of affinity with the viral 3'UTR): AeMaleless, an ATP-dependent helicase putatively involved in dosage compensation [30]; AeSex-lethal, an RNA-binding protein also putatively involved in dosage compensation [31]; an AeGTPase with putative function in translation regulation; AeStaufen, a dsRNA-binding protein that can induce RNA decay and transport [32]; AAEL001518, a protein with no homolog in human and an uncharacterized homolog in *Drosophila melanogaster*; AAEL004834, a putative nucleotide-binding protein; AeRan, a GTPase that is involved in RNA transport [33]; AeDISCO-interacting protein 1 (AeDIP1), a dsRNA-binding protein that is involved in antiviral defense [34]; AeRpS24, a ribosomal protein; AAEL014376, an uncharacterized protein; AeRNase, a ribonuclease protein; AeExoRNase, an exonuclease protein with homology to the *Drosophila* RNA-binding protein egalitarian [35]; AeAtu (AeAnother-Translation-Unit), a transcription-regulatory protein [36]; and AePur (AePurine-rich element binding protein), a DNA- and RNA-binding protein [37]. While most of the interacting RBPs have a RNA binding domain, these and the others could either directly bind to the 3'UTR or indirectly through a complex of proteins. As demonstrated previously in human [21], RBPs interacting with the 3'UTR of DENV2 may widely vary from RBPs interacting with the other DENV serotype 3'UTR.

## Both AePur and AeStaufen interact with DENV2 3'UTR, while AeStaufen also binds sfRNA

We were particularly interested in two 3'UTR-interacting proteins, AePur and AeStaufen. Since interaction to DENV2 RNA is conserved in the human homolog for AePur [18], the function of this protein could be conserved in both mosquitoes and humans. In the case of

**Table 1. List of *A. aegypti* proteins that interact with DENV2 3'UTR and their corresponding human homologs.**

| *Aedes aegypti* | | | Human homolog | Fold changes in Human screens for flavivirus RNA-binding proteins[2,3] | | | | | | |
| Aag2 –DENV2 3'UTR[1] | | | | Huh7 – DENV2 5'+ 3' UTR (1) | Huh7 – DENV1 3'UTR (2) | Huh7 – DENV2 3'UTR (2) | Huh7 – DENV3 3'UTR (2) | Huh7 – DENV4 3'UTR (2) | Huh7 – DENV2 gRNA (3) | Huh7 – ZIKV gRNA [11] |
| Gene ID | Gene name | Fold change | | | | | | | | |
|---|---|---|---|---|---|---|---|---|---|---|
| AAEL004859 | AeMaleless | 4.17 | DHX9; DHX29; DHX36; DHX57 | | 0 .55 | 0.54 1.30 | 0.34 | 0.65 | 3.43 1.46 1.39 | 2.48 1.35 1.78 1.61 |
| AAEL011150 | AeSex-lethal | 3.78 | ELAVL1; ELAVL2; ELAVL4; ELAVL3 | | | | | | 2.8 | 2.20 |
| AAEL003813 | AeGTPase | 2.30 | MTG1 | | | | | | | |
| AAEL019885 | AeStaufen | 2.24 | STAU1 | | 0.58 | 0.71 | 0.53 | 0.71 | 2.32 | 2.59 |
| AAEL001518 | AAEL001518 | 2.23 | Not identified | | | | | | | |
| AAEL004834 | AAEL004834 | 2.16 | JADE3 | | | | | | | |
| AAEL009287 | AeRan | 1.94 | RAN | 1.76 | | | | | 2.60 | |
| AAEL012964 | AeDIP1 | 1.92 | ADARB1 | | | | | | | |
| AAEL014292 | AeRpS24 | 1.92 | RpS24 | | | 0.68 | | 0.57 | | |
| AAEL014376 | AAEL014376 | 1.73 | C3orf17 | | | | | | | |
| AAEL001089 | AeRNase | 1.66 | RPP14 | | | 1.22 | | | | |
| AAEL002463 | AeExoRNase | 1.59 | EXD1 | | | | | | | |
| AAEL006172 | AeAtu | 1.55 | LEO1 | | | 1.22 | | | | |
| AAEL012134 | AePur | 1.54 | PURB | 1.94 | | | | | 2.04 | |

[1]Work carried out using RNA-affinity chromatography.

[2]Human studies using RNA-affinity chromatography or ChIRP.

[3]Cell type and viral fragments used are indicated.

AeStaufen, its putative function in RNA decay [32] could modulate the quantity of viral RNA fragments [24]. For these reasons, we validated our RNA-affinity chromatography results with RIP in infected condition for AePur and AeStaufen. RIP provided supportive and complementary information to the RNA-affinity chromatography by revealing *in vivo* interactions between proteins and virally-produced 3'UTR. Pur proteins are a family of single-stranded nucleic acid-binding proteins that are highly conserved from bacteria through humans [38]. Accordingly, we were able to use a commercially available antibody that targets human PurB and recognized AePur in *Ae. aegypti* mosquitoes (S1 Fig), to pull down AePur from lysates of orally-infected mosquitoes (viral blood titer at $10^6$ pfu/ml) (Fig 2A). Using primers that target the envelope gene located in the single open reading frame of DENV gRNA, we observed an enrichment for gRNA (4.59 ± 0.43 fold change) in AePur immunoprecipitates as compared to the IgG control (Fig 2B). Variation in enrichment between AePur and IgG IP indicated variation in affinity as we used the same input in both conditions. To determine whether AePur binds to gRNA 3'UTR or to sfRNA, both of which have the same sequence, we also quantified sfRNA enrichment as previously described [27]. SfRNA was not enriched in AePur immunoprecipitates as compared to the IgG control (Fig 2B). Together with the RNA-affinity chromatography results, these data validate AePur association with the 3'UTR sequence in DENV2 gRNA.

We next tried to confirm AeStaufen association with DENV2 3'UTR in infected mosquitoes, using the same approach as for AePur. However, attempts to immuno-precipitate AeStaufen using commercially available antibodies developed against its human homolog were

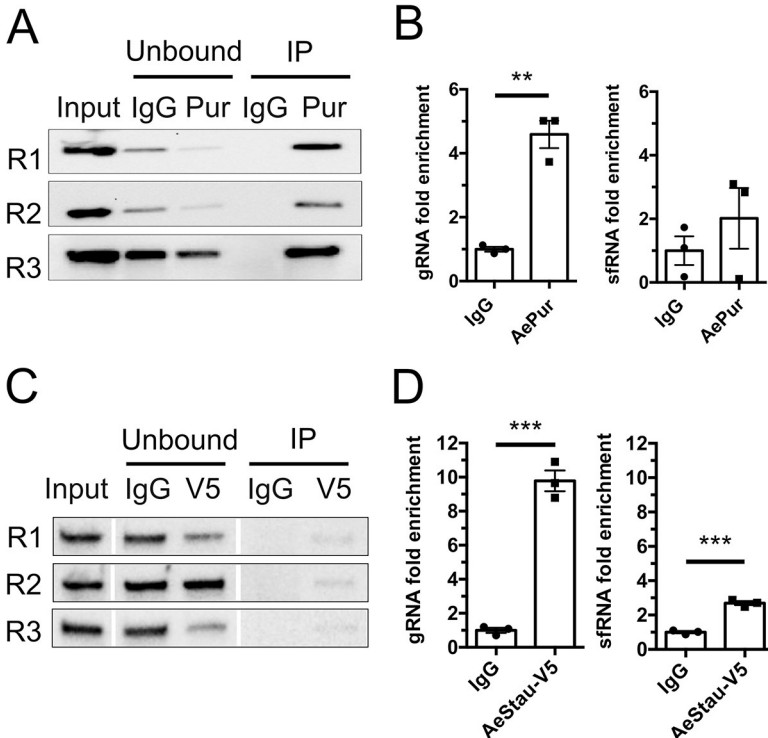

**Fig 2. Binding affinities of AePur and AeStaufen to DENV2 3'UTR and sfRNA. (A)** Western blots of AePur immunoprecipitates from orally-infected mosquito lysates. **(B)** gRNA and sfRNA fold enrichments in AePur immunoprecipitates as compared to IgG control. **(C)** Western blots of AeStaufen-V5 (AeStau-V5) immunoprecipitates from infected C6/36 cell lysates. **(D)** gRNA and sfRNA fold enrichments in AeStaufen immunoprecipitates as compared to IgG control. (A, C) Three biological repeats (R1-3) are shown. (B, D) Bars show mean ± s.e.m. from three repeats. **, p-value < 0.01; ***, p-value < 0.001 as determined by unpaired T-test.

unsuccessful. Therefore, we designed an alternative approach that uses C6/36 cells derived from *Aedes albopictus* mosquitoes, which are highly susceptible to DENV infection [39]. C6/36 cells were transfected with V5-tagged AeStaufen (S2 Fig) just before infecting them with DENV2. Forty-eight hours post-infection, cell lysates were subjected to RIP using an anti-V5 antibody or IgG control. While the AeStaufen-V5 bands were faint in the pull-downs, there was a clear depletion of the protein in the unbound fractions (Fig 2C), indicating a moderately efficient RIP. Nonetheless, DENV gRNA was enriched 9.78 ± 0.61 fold in V5 immunoprecipitates (Fig 2D). As done in AePur immunoprecipitation, we also determined the affinity of AeStaufen-V5 to sfRNA and observed a lower but significant enrichment (2.69 ± 0.12) for sfRNA in AeStaufen-V5 immuno-precipitates. Together, these results suggest that AeStaufen has a high affinity to gRNA 3'UTR and also interacts with sfRNA. Overall, using RIP experiments, we have validated the 3'UTR-protein interactions identified with the RNA-affinity chromatography and determined the affinity of AePur and AeStaufen for DENV2 3'UTR and sfRNA.

## Functional characterization of DENV2 3'UTR-bound proteins in mosquitoes

To test whether the mosquito proteins that associate with DENV2 3'UTR influence mosquito infection, we performed *in vivo* dsRNA-mediated silencing for each of the 14 proteins. As control, we injected a dsRNA targeting the bacterial gene *LacZ*. Four days post dsRNA-injection,

we quantified gene silencing in whole mosquitoes and observed a wide variation in silencing efficiency (S3 Fig). Variation in mRNA levels should be interpreted with caution as it does not directly correlate with protein levels and could either under- or over-estimate protein depletion. Nonetheless, in absence of antibodies for protein quantification, we considered that a decrease in mRNA associated with a change in phenotype indicated a successful protein depletion. DsRNA-injected mosquitoes were then infected by offering a blood meal containing $10^6$ DENV pfu/ml, which is within the range of inoculum concentration observed in patient serums [40]. The blood feeding rate was lower only in mosquitoes whose *AeMaleless*, *AeStaufen*, *AeRan* and *AePur* genes were silenced, while other silencing had no effect (S4A Fig).

Seven days post oral infection, we quantified DENV titers in whole mosquitoes using focus forming unit (FFU) assay. As higher infection can reduce mosquito survival [41] and introduce a bias towards the selection of mosquitoes surviving a lower infection, we evaluated the effect of silencing on survival. None of the silencing conditions influenced survival rate (S4B Fig). We then calculated infection rate as the percentage of mosquitoes with at least one viral particle over blood-engorged mosquitoes, and infection intensity as the number of FFUs per infected mosquito. The infection rate provides information about the ability of the virus to initiate infection, while infection intensity is a measure of virus multiplication. Nonetheless, these two biological parameters are interdependent; for instance, reduced viral multiplication (i.e. infection intensity) may lead to viral elimination that would then lower infection rate.

In terms of infection rate, we observed that silencing of *AeRan* and *AeExoRNase* reduced the percentage of infected mosquitoes by 23.5 and 29 points, respectively (Fig 3A), suggesting that these two proteins facilitate infection onset. In terms of infection intensity, four other proteins had an effect (Fig 3B). *AeRNase* silencing led to a 6.1-fold decrease in the number of FFUs per infected mosquito, indicating its function in facilitating virus multiplication. Inversely, FFUs per infected mosquito increased 10-fold, 15.7-fold and 4.2-fold upon the silencing of *AeGTPase*, *AeAtu* and *AePur*, respectively (Fig 3B), revealing anti-viral functions for these three proteins. Taken together, this *in vivo* mid-throughput screening provides the first evidence that RBPs influence DENV infection in the mosquito vector.

## The human homologs of AePur associates with DENV gRNA but do not alter infection

In humans, three genes (PURA, PURB, and PURG) encode for four Pur proteins (PurA, PurB and two isoforms of PurG). The highly-conserved purine-rich element binding domain (Pur domain) that is characteristic of Pur proteins and responsible for their interaction with nucleotides [38] is present in both the human and mosquito homologs (S5 Fig). To investigate whether interaction with DENV gRNA is conserved between AePur and HsPur proteins, we overexpressed Flag tagged-HsPurA and His-tagged HsPurB in human cells before infecting the cells with DENV2. We chose to study the HsPurA and HsPurB homologs since they are most closely related to AePur (S5 Fig). We used ectopic expression of tagged-proteins as available antibodies were not suitable for IP. Twenty-four hours post infection, we conducted RIP with either of the tags (Fig 4A) and quantified gRNA. DENV gRNA was enriched $150 \pm 3.8$ fold in HsPurA precipitates and $16.6 \pm 0.5$ fold in HsPurB precipitates as compared to the IgG control (Fig 4B). A significant part of the overexpressed proteins remained in the unbound fractions (Fig 4A), suggesting that gRNA-protein interactions were underestimated. These results show that association with the DENV genome is conserved across Pur homologs in both humans and mosquitoes.

We next tested if human Pur proteins influenced DENV2 infection. We separately depleted either HsPurA or HsPurB expression using siRNA-mediated silencing in human cells. To

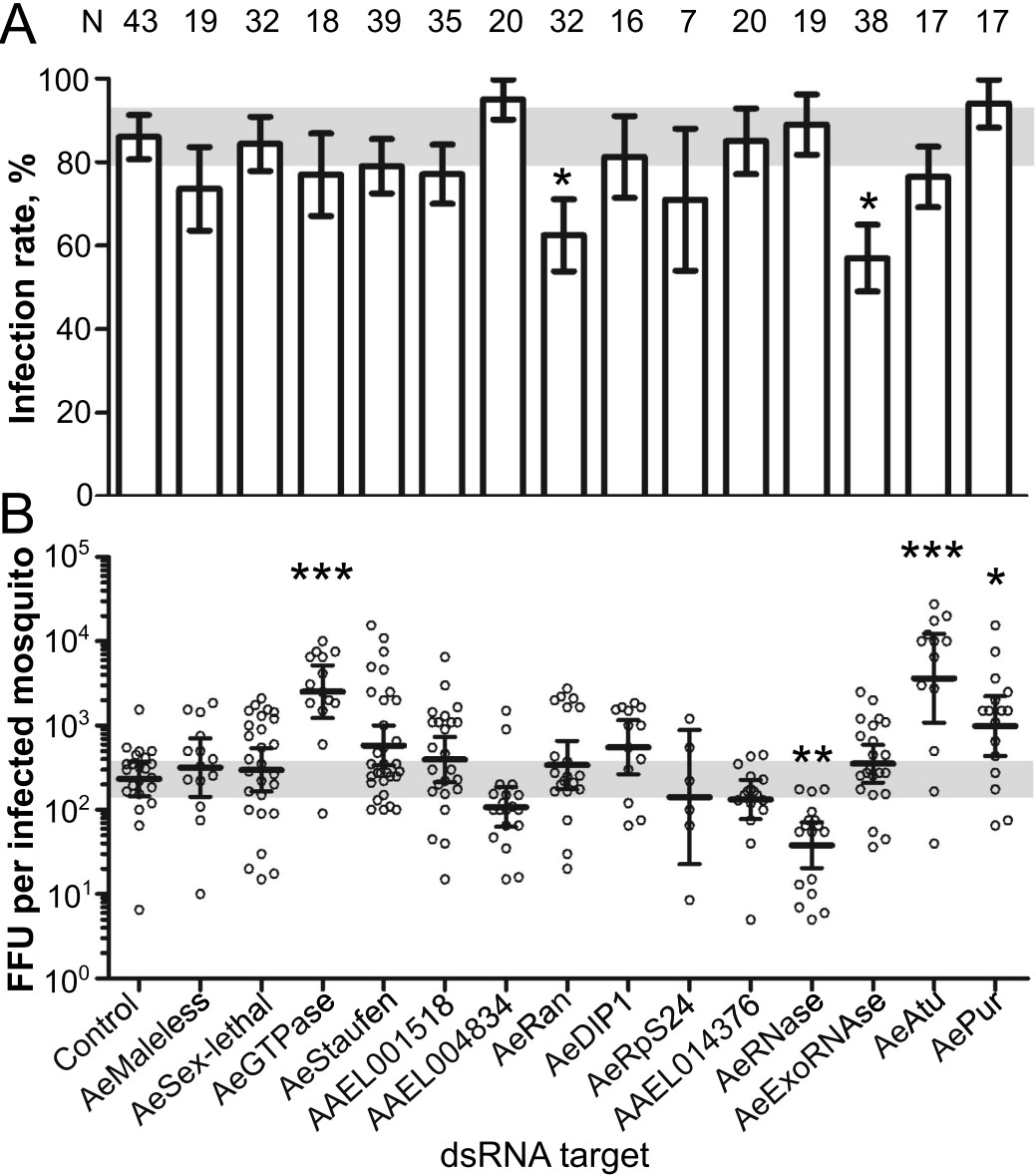

**Fig 3. Effects of the 3'UTR-bound proteins on DENV infection in mosquitoes.** Four days post-dsRNA injection to silence each of the 14 3'UTR-bound proteins, mosquitoes were orally infected with $10^6$ pfu/ml of DENV2. Infection was quantified 7 days later using focus forming unit (FFU) assay. **(A)** Infection rate. Bars show percentage ± s.e. *, p-value < 0.05 as determined by Z-test. **(B)** FFU per infected mosquito. Bars show geometric means ± 95% C.I. *, p-value < 0.05; ***, p-value < 0.001 as determined by Dunnett's test compared to control mosquitoes injected with dsRNA control. N, number of orally-infected mosquitoes. Data from three biological repeats, each using a specific set of dsRNA targets, were combined. DsRNA control was included in each repeat.

prevent untargeted silencing effects, we used three different siRNA sequences for each of the two Pur proteins and observed specific depletion of either HsPurA or HsPurB (Fig 4C). The depleted cells were infected with DENV2 and viral titer was quantified using plaque forming unit (PFU) assay 24h later. There was no effect of either HsPURA or HsPURB silencing on viral titer as compared to non-transfected (NT) and siRNA-transfected controls (siNC) (Fig 4D). To rule out the possibility that background levels of the HsPur proteins and/or functional

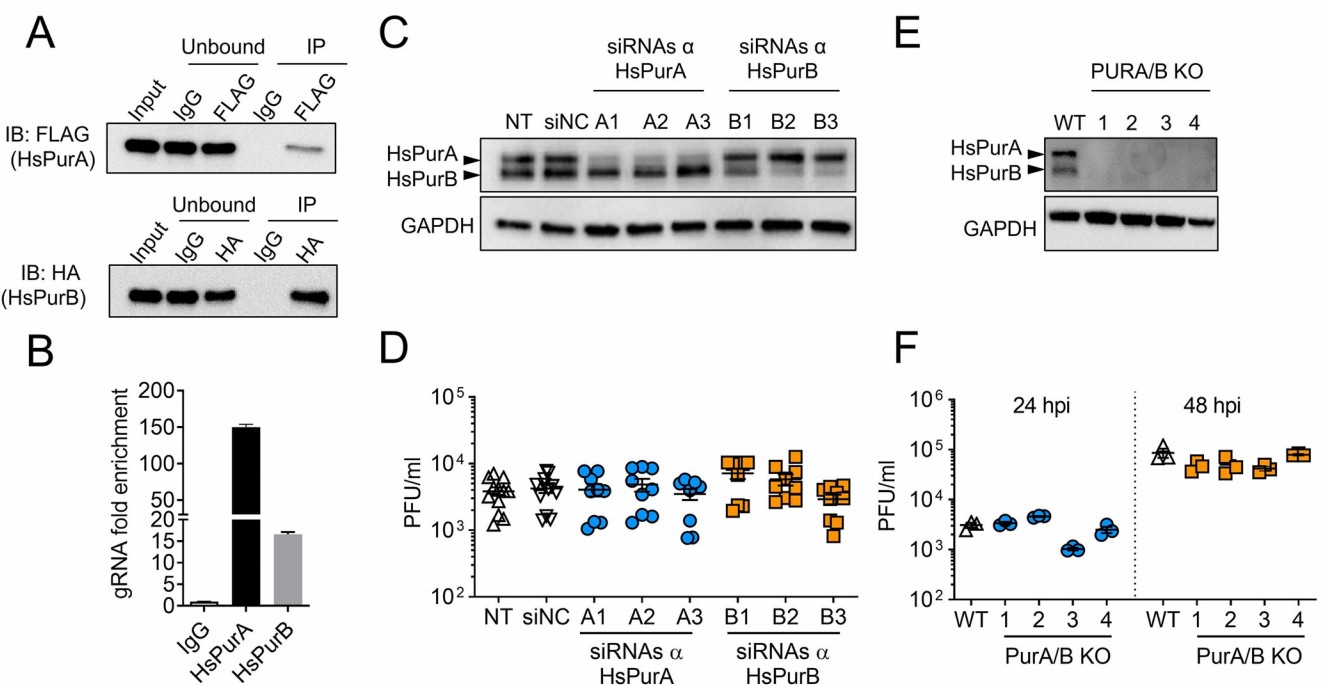

**Fig 4. Human PurA and PurB interactions with DENV RNA and effect on infection in human cells. (A)** Western blots of FLAG-tagged HsPurA and HA-tagged HsPurB immunoprecipitates from Huh7 cells 24 hpi with DENV2 at a MOI of 5. **(B)** gRNA and sfRNA fold enrichments in FLAG-tagged HsPurA and HA-tagged HsPurB immunoprecipitates as compared to IgG control. Bars show mean ± s.e.m. from two repeats. **(C)** Western blots of HsPurA and HsPurB from cells silenced for either of the HsPUR proteins. Three different siRNAs were used for each protein. The picture is representative of multiple repeats. GAPDH was used as a loading control. **(D)** Effect of *HsPURA* or *HsPURB* silencing on viral load estimated by plaque forming unit (PFU) quantities in supernatants at 24 hpi with a MOI of 0.1. Bars indicate means ± s.e.m. **(E)** Western blots of HsPurA and HsPurB from cells, where both proteins were genetically-depleted. Four different double knock-out lines (PURA/B KO) for each protein were tested. The picture is representative of multiple repeats. GAPDH was used as a loading control. **(F)** Effect of HsPurA and HsPurB genetic ablation on viral loads estimated by PFU quantities in supernatants at 24 and 48hpi, with a MOI of 0.1. Bars indicate means ± s.e.m. NT, non-transfected control; siNC, transfected with control siRNA; WT, wild-type.

redundancy between HsPurA and HsPurB could be responsible for unaffected DENV titers, we generated cells devoid of both HsPurA and HsPurB via CRISPR/Cas9 editing. To prevent untargeted effects of the knock-out approach, we produced four different cell lines in which both HsPurA and HsPurB expressions were abrogated (Fig 4E). Similar to the silencing approach (Fig 4D), DENV titers remained unaffected in the absence of both HsPur proteins (Fig 4F). Altogether, these results indicate that HsPurA and HsPurB strongly interact with DENV2 gRNA like AePur. However, although we have not tested the function of HsPurG, our results suggest that the anti-viral function of AePur is not conserved in the human homologs.

## AeStaufen reduces gRNA and sfRNA quantities in mosquito carcass, midgut and salivary glands but does not affect viral titer

To functionally characterize AeStaufen, we first quantified its relative expression under non-infected and infected conditions in whole mosquito, midgut, salivary glands, and carcass (remains after midgut and salivary glands dissection). *AeStaufen* expression was normalized to that of *Actin* for comparison among the tissues. While *AeStaufen* expression was not affected by oral infection within each of the tissues, salivary glands had the highest *AeStaufen* expression among the tissues (Fig 5A). *AeStaufen* expression was also high in the carcass, suggesting that organs remaining in the carcass, such as ovaries and brain, express high levels of *AeStaufen* as observed in *D. melanogaster* [42]. The expression pattern of *AeStaufen* indicates its

ubiquitous presence, with important functions in certain tissues, such as the salivary glands, where it is most expressed.

To evaluate the effect of AeStaufen on DENV2 infection in the carcass, midgut and salivary glands, we silenced *AeStaufen* by injecting a large quantity of dsRNA that is sufficient to decrease mRNA levels across the mosquito organs (S6 Fig). A similar amount of *dsLacZ* was injected as control. The mosquitoes were orally infected (viral titer of $10^7$ pfu/ml) at four days post-dsRNA injection, and gRNA was quantified at 10 days post-oral infection (dpi) in the carcass, midgut and salivary glands. To exclude bias caused by enhanced survival of less-infected mosquitoes, we evaluated blood feeding and survival rates upon AeStaufen depletion and did not report any effect of the silencing (S2 Table). Similar to FFU quantification described above, we calculated infection rate as the percentage of tissues with at least one gRNA and infection intensity as the number of gRNA copies per infected tissue. Infection rates in all the tissues were unaffected by *AeStaufen* silencing, suggesting that AeStaufen does not influence infection onset (Fig 5B). However, infection intensity increased 3.02-fold and 5.67-fold upon AeStaufen depletion in the carcass and salivary glands, respectively (Fig 5B). It is interesting to note that these tissues exhibited the highest *AeStaufen* expression (Fig 5A). In the midgut, AeStaufen depletion led to a 3.61-fold increase in infection intensity (Fig 5B), although the difference was not statistically significant (p = 0.12, as determined by t-test). These results reveal that AeStaufen mediates reduction in DENV gRNA copies in multiple organs of mosquitoes.

Since AeStaufen also interacts with DENV2 sfRNA (Fig 2D), we then determined the effect of AeStaufen depletion on sfRNA copies in the same samples. We calculated the sfRNA detection rate to inform about the initiation of sfRNA production, and the sfRNA copies per infected tissue to evaluate the effect on the production and/or degradation of sfRNA. While AeStaufen depletion did not change sfRNA detection rate, it increased sfRNA copies per infected tissue by 6-fold, 14.43-fold, and 10.99-fold in the carcass, midgut, and salivary glands, respectively (Fig 5C). To normalize sfRNA quantity to the amount of its precursor and the level of infection (both determined by gRNA), we calculated the sfRNA:gRNA ratio. AeStaufen depletion increased the sfRNA:gRNA ratio only in the midgut and salivary glands, by 2.19-fold and 2.47-fold, respectively (Fig 5D). Altogether, these results indicate that AeStaufen mediates decrease in both gRNA and sfRNA copies in all tissues, but that its effect on sfRNA is more pronounced in the midgut and salivary glands.

We have previously shown that DENV2 strains producing higher sfRNA copies in the salivary glands result in a higher viral titer in the tissue and a higher saliva-mediated infection rate [27]. To test whether higher sfRNA:gRNA ratio in the salivary glands resulting from AeStaufen depletion reproduces these observations, we quantified viral titers in the salivary glands and saliva of AeStaufen-depleted mosquitoes. To exclude bias caused by unsalivating mosquitoes, we measured the salivation rate and reported that it was not affected by *AeStaufen* silencing (S2 Table). Both infection rate and infection intensity were unaffected by AeStaufen depletion in the salivary glands and saliva (Fig 5E and 5F). Together with the lack of effect on infection in AeStaufen-depleted mosquitoes (Fig 3), these results indicate that AeStaufen does not alter the production of infectious viral particles.

## AeStaufen reduces the amount of sfRNA secreted in saliva

Our group had previously demonstrated that DENV secretes the anti-immune sfRNA in mosquito saliva to enhance saliva infectivity in human skin cells, thereby increasing viral transmission [28]. To determine whether AeStaufen's effect on sfRNA and gRNA in the salivary glands modifies the sfRNA:gRNA ratio in the saliva, we quantified gRNA and sfRNA in AeStaufen-depleted mosquito saliva at 10 dpi. gRNA detection rate and gRNA copies per infected saliva

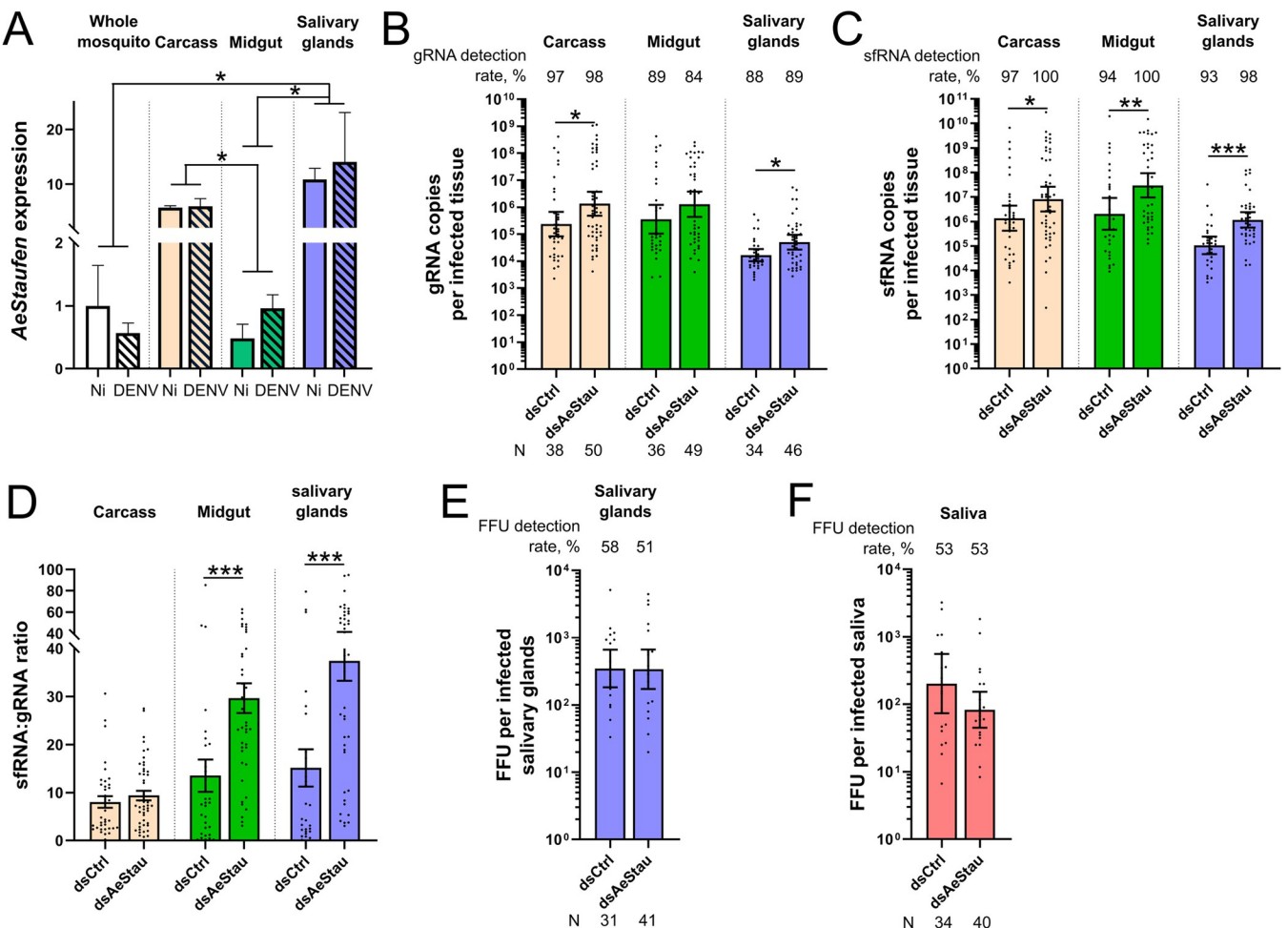

**Fig 5. Effect of AeStaufen on gRNA and sfRNA copies, and viral titers in different mosquito tissues. (A)** *AeStaufen* relative expression in whole mosquitoes, carcass (what is left after dissection), midgut, and salivary glands at 10 days post non-infectious (Ni) or infectious blood-feeding (DENV). Bars represent means ± s.e.m. from three replicates of pools of ten tissues. **(B-D)** Effect of *AeStaufen* silencing (*dsAeStau*) on gRNA copies (B), sfRNA copies (C) and sfRNA:gRNA ratio (D) in the carcass, midgut and salivary glands at 10 days post-oral infection. **(E-F)** Effect *AeStaufen* silencing (*dsAeStau*) on viral titers as determined by focus forming unit (FFU) assay in salivary glands (E) and saliva (F) at 10 days post-oral infection. (B-F) *dsLacZ* was injected as control. (B, C, E, F) Bars indicate geometric means ± 95% C.I. (D) Bars indicate means ± s.e.m. N, number of mosquitoes analyzed. *, p-value < 0.05; **, p-value < 0.01; ***, p-value < 0.001 as determined by unpaired t-test or by LSD-Fisher test (A).

were very similar in AeStaufen-depleted and control mosquitoes (Fig 6A). Inversely, sfRNA:gRNA ratio increased 1.89-fold upon AeStaufen depletion, although sfRNA detection rate was unaltered (Fig 6B).

We next tested whether the increase in sfRNA:gRNA ratio in the saliva was caused by AeStaufen-mediated sfRNA degradation in the salivary glands. To infect salivary glands without going through midgut infection, we inoculated AeStaufen-depleted mosquitoes and quantified gRNA and sfRNA in salivary glands at 7 days post inoculation. gRNA detection rate (p = 0.10, as determined by Z-test) and gRNA copies per infected salivary glands were not altered (Fig 6C). However, similarly to what we observed in salivary glands from orally-infected mosquitoes, sfRNA:gRNA ratio was increased 1.77-fold in salivary glands from inoculated mosquitoes (Fig 6D). Altogether, these results indicate that AeStaufen-mediated reduction of sfRNA quantity in salivary glands influences the ratio of sfRNA:gRNA in saliva.

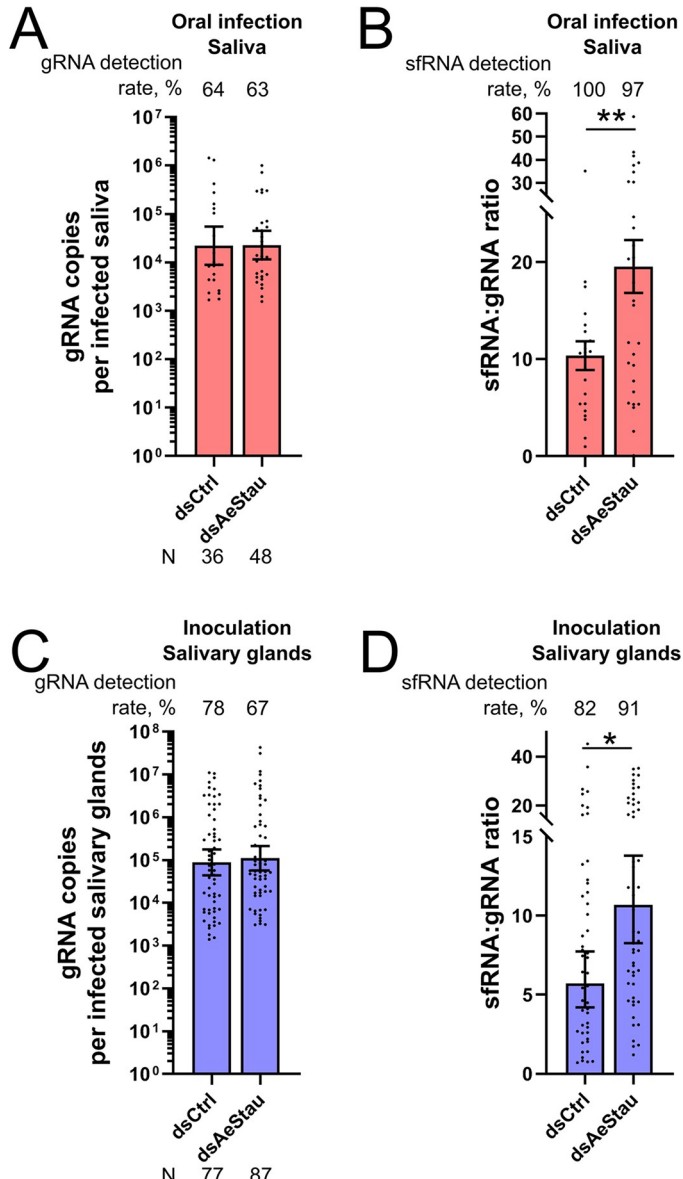

**Fig 6. Effect of AeStaufen on sfRNA secretion in the saliva. (A-B)** Effect of *AeStaufen* silencing (*dsAeStau*) on gRNA copies (A) and sfRNA:gRNA ratio (B) in saliva collected at 10 days post-oral infection. **(C-D)** Effect of *AeStaufen* silencing (*dsAeStau*) on gRNA copies (C) and sfRNA:gRNA ratio (D) in the salivary glands at 7 days post-inoculation. *dsLacZ* was injected as control. (A, C) Bars indicate geometric means ± 95% C.I. (B, D) Bars indicate means ± s.e.m. N, number of mosquitoes analyzed. *, p-value < 0.05; **, p-value < 0.01 as determined by unpaired t-test.

## Discussion

While the importance of RBPs in DENV life cycle is well-supported in mammals [7], there is a dearth of knowledge about RBP identities and functions in mosquitoes. By successfully pioneering the use of RNA-affinity chromatography in mosquito cells, we discovered fourteen mosquito RBPs that interact with DENV2 3'UTR in *Ae. aegypti*. Furthermore, using *in vivo* functional analyses, we determined that three RBPs have pro-viral functions and three others have anti-viral functions. We then evaluated the affinities of two RBPs, AePur and AeStaufen,

for either 3'UTR or sfRNA (as both share the same sequence), concomitantly validating the RNA-affinity chromatography results. AePur interacts specifically with the 3'UTR, whereas AeStaufen associates with both 3'UTR and sfRNA. We further showed that the affinity for DENV RNA is conserved across AePur homologs, although only the mosquito homolog has anti-viral functions. Importantly, we revealed that AeStaufen mediates reduction of gRNA in the carcass and salivary glands, and of sfRNA in the carcass, midgut, and salivary glands. While AeStaufen-mediated decay of viral RNA does not influence the number of infectious viral particles, it reduces the concentration of transmission-enhancing sfRNA in the saliva, revealing AeStaufen impact on viral transmission. By characterizing the first DENV RBPs in mosquitoes, this study unravels their multipronged functions in DENV transmission.

Interestingly, the affinity to DENV RNA is conserved in the human homologs of eight of the 14 RBPs we identified in mosquitoes (Table 1). A Comprehensive identification of RNA-binding proteins by mass spectrometry (ChIRP-MS) study in human cells found that DHX9, DHX29, DHX36, and DHX57 (homologs of AeMaleless) and ELAVL1 (AeSex lethal homolog) interact with DENV2 and Zika virus genomes [11]. However, an RNA-affinity chromatography approach (similar to the one used in this study) found that DHX9 has less affinity to the 3'UTR of DENV1-4 than to a control RNA fragment, while DHX36 only marginally (1.30-fold) interacts with DENV2 3'UTR [43]. These contrasting results for genomic RNA and 3'UTR affinities suggest that the RBPs preferentially interact with sequences outside the 3'UTR in human cells. Furthermore, Staufen 1 (AeStaufen homolog) associates with DENV2 and Zika virus gRNA [11] but not the 3'UTR of DENV1-4 [21]. Ran (AeRan homolog) interacts with a concatenated sequence of DENV2 5'UTR and 3'UTR as identified by RNA-affinity chromatography [18], and with the DENV2 genome as identified by ChIRP-MS [11]. RPP14 (AeRNase homolog) and LEOA (AeAtu homolog) interact with DENV2 3'UTR [21], while PURB (AePur homolog) associates with the concatenated sequence of 5' and 3'UTRs [18]. The conservation of RBP affinity for DENV RNA across human and mosquito homologs suggests that the virus has evolved to exploit the biological similarities between its two hosts.

We have identified six RBPs that influence DENV infection in mosquitoes. We found that AeRan, a small GTPase, favors DENV infection by promoting the infection rate. Its human homolog, Ran, has a well-supported function in the nucleocytoplasmic transport of RNA [44]. Contrary to its pro-DENV effect in mosquitoes, *D. melanogaster* Ran contributes to anti-viral immune response by regulating virus phagocytosis and enabling the nuclear translocation of transcription factors activated by the Toll signaling pathway [45,46]. We also found that AeExoRNase (a predicted exoribonuclease) is required for optimal infection rate in mosquitoes. Its *Drosophila* homolog, egalitarian, has RNA-binding capacity and is involved in RNA localization [47,48], suggesting that AeExoRNase is required for proper DENV gRNA transport. Moreover, we observed that AeRNase (a subunit of ribonuclease P) enhances infection intensity in mosquitoes, indicating a putative new role for ribonuclease P in virus multiplication. Inversely, an AeGTPase with no functional data for its homologs reduces infection intensity. AeAtu, whose homologs in *D. melanogaster* (i.e. Atu) and humans (i.e. LEO1) are involved in gene regulation [36], had the strongest anti-viral impact. LEO1 is a component of Polymerase Associated Factor 1 complex (PAF1C), which mediates a strong anti-viral response through gene regulation [49]. Although AeAtu function in mosquitoes is unknown, it is tempting to speculate that the PAF1C immune regulation is conserved in mosquitoes. Finally, we noted that AePur, an RNA- and DNA-binding protein, mediated a reduction in infection intensity. The role of one of the human homologs, PurA, in promoting stress granule formation [50] that is detrimental to viral infection [51] may provide hints on how AePur regulates infection intensity. However, while the binding to DENV RNA was conserved across the human and mosquito Pur homologs, the anti-viral effect was specific to the mosquito protein, indicating

functional divergence between the homologs with respect to DENV infection. Together, our mid-throughput *in vivo* screening is the first study to identify RBPs that influence viral multiplication in mosquitoes, and highlights potential new targets for blocking viral transmission.

Importantly, we have revealed that AeStaufen mediates a reduction in both DENV gRNA and sfRNA copies. The human and *Drosophila* Staufen homologs bind dsRNA through five dsRNA-binding domains [52], which are conserved in AeStaufen (S7 Fig). SfRNA contains several dsRNA sequences [53] and gRNA possesses multiple hairpins in highly structured regions [54], both of which could allow AeStaufen interaction. Upon binding, Staufen can transport nucleotides to ribonucleoprotein complexes that modulate translation. Alternatively, Staufen can initiate the assembly of an mRNA decay complex, called Staufen-mediated RNA decay (SMD) [32]. Since both its functions are associated with mRNA processing, Staufen is thought to act as post-transcriptional regulator. Altogether, we propose that the increase in gRNA copies that is observed upon AeStaufen depletion results from the inhibition of the SMD pathway in DENV-infected mosquitoes.

While several viruses rely on Staufen to complete their life cycles [55–57], this is the first evidence that a Staufen homolog hinders virus multiplication by degrading its gRNA. Another major RNA decay process, called nonsense mRNA decay (NMD), was previously shown to degrade viral RNA and reduce viral titer [58]. However, in our study, the increase in gRNA copies upon AeStaufen depletion did not translate into higher number of infectious particles in both whole mosquitoes and salivary glands. Such discrepancy between gRNA copies and virion number suggests that the cellular localization of gRNA that is degraded is different from that of the gRNA that is assembled in virions. Positive-sense gRNA is produced in replication complexes located at the endoplasmic reticulum (ER) and is then assembled at a distinct ER assembly site [7,59]. The translocation of newly-synthesized viral RNAs from the replication to assembly site is aided by viral and host cellular RBPs [59,60]. Nevertheless, AeStaufen may be degrading gRNA that is not directed to the assembly sites, but is instead transported to other cellular compartments. Staufen is usually found in stress granules, where the SMD machinery is assembled [61]. We hypothesize that the gRNA molecules that are not directed towards virion production sites are released actively or passively from the replication complexes, where they are degraded by an AeStaufen-mediated mechanism.

SfRNA, which is derived from gRNA degradation, was increased upon AeStaufen depletion in the carcass, midgut and salivary glands. Increased sfRNA production could result from a higher quantity and availability of its gRNA template. In this case, higher gRNA copies as reported in carcass and salivary glands would proportionally increase sfRNA copies independently of AeStaufen. To inform about a proportional relationship between sfRNA and gRNA quantities, we calculated sfRNA:gRNA ratio. In the carcass, the ratio was unchanged by AeStaufen depletion, suggesting that it is the higher gRNA copies that led to the increased production of sfRNA. However, higher sfRNA:gRNA ratios in AeStaufen-depleted midgut and salivary glands imply that gRNA and sfRNA quantities are uncoupled (ie., sfRNA copies are not solely dependent on the amount of gRNA copies). Together with the binding of AeStaufen to sfRNA, these results indicate that AeStaufen mediates sfRNA degradation in the midgut and salivary glands.

SfRNA has multipronged functions in viral transmission [24]. We had previously reported that higher sfRNA concentration in the salivary glands increases virion quantity by inhibiting the Toll immune pathway [27]. Surprisingly, a higher sfRNA concentration in the salivary glands induced by AeStaufen depletion did not result in increased virus titer in the salivary glands and saliva. As for gRNA, this may stem from different cellular localizations of the sfRNA molecules degraded via AeStaufen and the sfRNA molecules involved in regulating immunity. The Toll pathway components are localized in plasma membrane or cytosol [62],

whereas SMD complexes are localized in stress granules [61]. We also previously reported that sfRNA is secreted in salivary extracellular vesicles to enhance infection at the bite site [28]. Here, we have shown that AeStaufen-mediated reduction of sfRNA takes place in the salivary glands and that this reduces the quantity of secreted sfRNA. Altogether, these results show that AeStaufen regulates viral transmission by modulating salivary sfRNA concentrations.

In conclusion, our successful adaptation of a biochemical technique employed in mammals has led to the identification of multiple mosquito RBPs that associate with DENV RNA and modulate its life cycle. Since several of these RBPs have roles in viral multiplication, they could be used as potential targets for the design of novel, effective interventions for dengue prevention. Furthermore, by describing the effects of AeStaufen on DENV gRNA and sfRNA levels, we have unraveled a new role for AeStaufen in viral transmission by mosquitoes. Our identification of the multiple ways RBPs can regulate DENV transmission motivate further studies of RBPs in mosquitoes.

## Materials and methods

### Cell lines, virus, and mosquitoes

*Aedes albopictus* C6/36 (CRL-1660) and baby hamster kidney BHK-21 (CCL-10) cell lines obtained from ATCC, and *Aedes aegypti* Aag2 cells received from Dorothée Missé's lab were grown in Roswell Park Memorial Institute medium (RPMI, Gibco). Human hepatic Huh7 cells (JCRB0403) were maintained in Dulbecco's Modified Eagle Medium (DMEM, Gibco). For all cell lines, the medium was supplemented with 10% heat-inactivated fetal bovine serum (FBS) (ThermoFisher Scientific), 100 U/ml penicillin, and 100 μg/ml streptomycin (Thermo-Fisher Scientific). Mosquito cells were cultured at 28˚C with 5% $CO_2$, and mammalian cells were grown at 37˚C with 5% $CO_2$.

Dengue virus 2 (DENV2) New Guinea C (NGC) strain from ATCC (VR-1584) was propagated in C6/36 cells and titrated using plaque assay (this determines the number of plaque forming units, pfu) in BHK-21 as previously detailed [27].

The *Aedes aegypti* colony was established from eggs collected in Singapore in 2010 and reared in the insectary thereafter. The eggs were hatched in MilliQ water and the larvae were kept at a density of 2.5–3 larvae/$cm^2$ in shallow water and fed on a mixture of Tetra-Min fish flakes (Tetra), yeast, and liver powder (MP Biomedicals). Adult mosquitoes were maintained in a 30×30×30 cm cage (Bioquip) and fed with 10% sucrose solution (1st base) *ad libitum*. They were maintained at 28˚C and 50% relative humidity in a 12h:12h light: dark cycle.

### RNA-affinity chromatography

The method was modified from [18] except that the Stable Isotope Labeling with Amino acids in Cell culture (SILAC) technique was not used. In brief, DENV2 3'UTR and size-matched NS2 control templates were amplified from viral cDNA using AW005 5'-CGGGTATGTGCG TCTGGATCCTATAAGAAGAGGAAGAGGCAGG-3' and AW043 5'-AGAACCTGTTGAT TCAACAGCAC-3', and AW024 5'-CGGGTATGTGCGTCTGGATCCTATGCAGCTGGAC TACTCTTGAG-3' and AW047 5'-GGTCCTGTCATGGGAATGTC-3', respectively. A T7-flanked tobramycin aptamer was incorporated at the 5'-end of the templates. RNA was generated using MegaScript T7 transcription kit (Invitrogen), folded by heating and subsequent slow cooling, and bound to a tobramycin bead matrix. Beads decorated with 3'UTR or NS2 fragments were incubated with the same amount of pre-cleared lysate from Aag2 cells. Beads were washed and proteins were eluted along with excess tobramycin using MicroSpin columns (Pierce). The experiment was repeated thrice using different batches of Aag2 cells.

## Quantitative mass spectrometry

RNA affinity chromatography eluates were analyzed by Data-Dependent Acquisition (DDA) quantitative MS. Trypsin-digested samples were first injected into a trap column (300 μm i.d. x 5 mm, C18 PepMap 100), and then into a C18 reversed-phase home-packed 15 cm column (SB-C18, ZORBAX, 5 micron, Agilent). Flow rate was maintained at 400 nL/min for a 60-min LC gradient, where mobile phase included A (5% ACN, 0.1% FA, Burdick and Jackson) and B (100% ACN, 0.1% FA). The eluted samples were sprayed through a charged emitter tip (PicoTip Emitter, New Objective, 10 ± 1 μm) into Orbitrap Fusion MS system (Thermo Fisher Scientific), coupled with a Dionex Ultimate 3000 nano HPLC (Thermo Fisher Scientific). The following parameters were used: tip voltage at +2.2 kV, FTMS mode for MS acquisition of precursor ions (resolution 120,000), and ITMS mode for subsequent MS/MS of top 10 precursors selected; MS/MS was accomplished via collision-induced dissociation (CID). The Proteome Discoverer 1.4 software was used for protein identification from *Ae. aegypti* UniProt entries, under the following parameters: maximum missed cleavages = 2; precursor tolerance = 5ppm; MS fragment tolerance = 0.6 Da; peptide charges considered = +2, +3, and +4. The significance of a peptide match was based on expectation values smaller than 0.05. Proteins were considered enriched in 3'UTR when present more than 1.5 fold than in NS2 control sequence with a p-value < 0.1 as determined by t-test after Bonferroni adjustment. The putative functions of the proteins were inferred from functional information available for *Drosophila* and human homologs.

## Mosquito oral infection

Three- to five-day-old female mosquitoes were starved for 24 h and offered a blood meal containing 40% volume of washed erythrocytes from specific-pathogen-free (SPF) pig's blood (PWG Genetics), 5% of 10 mM ATP (Thermo Fisher Scientific), 5% of human serum (Sigma) and 50% of RPMI medium containing DENV2 NGC. A blood viral titer of $10^6$ pfu/ml was used for the functional characterization of the 14 proteins interacting with DENV 3'UTR and for AePur RIP. Alternatively, a blood viral titer of $10^7$ pfu/ml was used for the functional characterization of AeStaufen. Blood viral titers were validated by plaque assay as described above. Mosquitoes were let to blood-feed for 1.5 h in the Hemotek membrane feeder system (Discovery Workshops) covered with porcine intestine (sausage casing). The blood meal titer may have decreased after 1.5 h. Fully engorged mosquitoes were selected and maintained in similar conditions as for the colony, with *ad libitum* access to water and 10% sugar solution. Blood-feeding rate was calculated by dividing the number of engorged mosquitoes by the total number of mosquitoes that were offered the blood meals. Survival rate was calculated at the collection time by dividing the number of living mosquitoes to the number of blood-fed mosquitoes.

## RNA immunoprecipitation for AePur (AAEL012134) and AeStaufen (AAEL007470)

For AePur RIP, mosquitoes were orally infected by feeding them blood containing $10^6$ PFU/ml DENV as described above. At 7 days post-infection (dpi), 10 mosquitoes were cold-anesthetized and homogenized in RIP lysis buffer [200 mM KCl, 20 mM HEPES pH7.2, 2% N-dodecyl-β-D -maltoside, 1% Igepal, 100 U/mL Murine RNase inhibitor (NEB)] using a bead mill homogenizer (FastPrep-24, MP Biomedicals). Homogenates were kept on ice for 30 min and centrifuged at 13,000 rpm for 15 min at 4˚C. Cleared lysates were sonicated in an ultrasound bath cleaner (JP Selecta Ultrasons system, 40 kHz) for 15 sec and placed on ice for 15 sec. The sonication procedure was repeated three times.

To perform AeStaufen RIP, *AeStaufen* cDNA synthesized by Genscript was cloned into pIZT/V5-His (Invitrogen). The same plasmid containing *Chloramphenicol acetyltransferase* (*CAT*) that was provided in the kit was used as expression control (S2A Fig). Four μg of either plasmid were transfected into 6 x 10$^5$ C6/36 cells using TransIT-2020 transfection reagent (Mirus) for 4 h. After washing, cells were infected with DENV2 at multiplicity of infection (MOI) = 5. Two days post-infection, cells were collected in 150 μl of lysis buffer [200 mM KCl (Sigma), 20 mM HEPES (pH 7.2) (Sigma), 2% N-dodecyl-β-D-maltoside (Thermo Fisher scientific), 1% Igepal CA-630 (Sigma), 100 U/mL Murine RNase inhibitor (NEB) and 1 X protease inhibitor cocktail (Roche)].

For RIP, 500 μg of protein lysate were diluted in 500 μl of NT2 buffer [50 mM Tris HCl (pH 7.4), 150 mM NaCl (Sigma), 1 mM $MgCl_2$ (Sigma), and 0.01% Igepal CA-630] and incubated with 5 μg of rabbit anti-V5 antibody (Sigma), anti-Pur (Bethyl Laboratories), or normal rabbit IgG (Merck) at 4˚C overnight. Bound complexes were captured into 50 μl of Dynabeads-protein G (Thermo Fisher Scientific) by rotation for 2 h at 4˚C and washed four times in NT2 buffer. Immunoprecipitates were analyzed by western blot and gRNA and sfRNA were quantified by RT-qPCR.

## Western blot

Immuno-precipitates or cells were lysed in RIPA lysis buffer (Cell Signaling Technology). Proteins were separated under denaturing conditions on 4–15% polyacrylamide gels (Bio-Rad) and transferred onto polyvinylidene difluoride membranes (PVDF, Bio-Rad). The membranes were blocked in 5% slim milk (Bio-Rad), diluted in PBS-T (1$^{st}$ Base), at room temperature for 30 min, and incubated with 1: 5,000 rabbit anti-V5 (Sigma), 1:1,000 rabbit anti-PurB (Bethyl Laboratories), 1:2,000 rat anti-PurA/PurB (kind gift from Robert Kelm, University of Vermont), 1:10,000 mouse anti-Actin (MA5-11869, Thermo Fisher Scientific) or 1:1,000 anti-GAPDH antibodies at 4˚C overnight. The blots were washed three times with PBS-T buffer, incubated with goat anti-rabbit HRP, goat anti-mouse HRP or goat anti-rat HRP (Jackson ImmunoResearch) at room temperature for 1 h. After three PBS-T washes, blots were visualized using chemiluminescence imaging system (Bio-Rad) with SuperSignal West (Thermo Fisher Scientific).

## Quantification of sfRNA and gRNA copies by real-time RT-qPCR

Mosquito tissues were homogenized in 350 μl of TRK lysis buffer [E.Z.N.A. Total RNA kit I (OMEGA Bio-Tek)] with silica beads (BioSpec) using mini-beadbeater (BioSpec). Saliva-containing mixture and cells were lysed in 350 μl of TRK lysis buffer without bead homogenization. Total RNA was extracted using the E.Z.N.A. Total RNA kit I protocol and eluted with 30 μl DEPC-treated water (Ambion).

gRNA was quantified by RT-qPCR using the iTaq Universal probe one-step kit (Bio-Rad) with primers and probe targeting the DENV2 envelope [27]. The 12.5 μl reaction mix contained 400 nM of forward and reverse primers, 200 nM of probe and 4 μl of RNA extract. sfRNA and 3'UTR were quantified together by RT-qPCR using the iTaq Universal Sybr green one-step kit (Bio-Rad) with primers previously designed [26]. The 10 μl reaction mix contained 300 nM of forward and reverse primers and 4 μl of RNA extract. Quantification was conducted with a CFX96 Touch Real-Time PCR Detection System (Bio-Rad). The thermal profile for gRNA quantification was 50˚C for 10 min, 95˚C for 1 min and 40 cycles of 95˚C for 10 sec and 60˚C for 15 sec, while that for sfRNA1/3'UTR quantification was 50˚C for 20 min, 95˚C for 1 min and 40 cycles of 95˚C for 10 sec and 60˚C for 15 sec, followed by melting curve analysis.

To absolutely quantify gRNA and sfRNA/3'UTR, we amplified templates encompassing either the gRNA or the sfRNA/3'UTR targets using forward T7-tagged primers; for gRNA we used 5'-CAGGATAAGAGGTTCGTCTG-3' and 5'-TTGACTCTTGTTTATCCGCT-3', resulting in a 453 bp fragment; for sfRNA we used 5'-AGAAGAGGAAGAGGCAGGA-3' and 5'-CATTGTTGCTGCGATTTGT-3', resulting in a 319 bp fragment. Templates were transcribed using the MegaScript T7 kit and purified using the E.Z.N.A total RNA extraction kit. As we did not treat the template samples with DNase, there may be some DNA left after RNA purification. Both DNA and RNA were simultaneously quantified at 260 nm absorbance using a NanoDrop 2000 spectrophotometer (ThermoFisher Scientific). Dilutions of specific quantities of RNA fragments were used to generate absolute standard curves.

sfRNA copy number was calculated by subtracting the number of gRNA fragments (estimated using the envelope primers) from the combined number of sfRNA and 3'UTR copies (estimated using the 3'UTR/sfRNA1 primers). For infected samples that contained detectable amount of sfRNA, sfRNA: gRNA ratio was calculated by dividing the number of sfRNAs over the number of gRNAs.

## Double-stranded RNA-mediated RNAi

The DNA templates used to generate dsRNA against the 14 3'UTR-interacting proteins were synthesized using T7-tagged primer pairs as detailed in S3 Table. A size-matched fragment from *E. coli LacZ* was used as the template for generating dsRNA control (dsCtl). The PCR fragments were transcribed using the MegaScript T7 transcription kit and the resulting dsRNA was purified using the E.Z.N.A. Total RNA kit I (Omega) and folded by heating to 95˚C for 2 min and slowly cooling down at 0.1˚C per second using a thermocycler (Bio-Rad). DsRNA concentrations were adjusted to 3 or 14 μg / μl using SpeedVac Concentrator (Thermo Scientific) and their integrity was checked on agarose gels. The dsRNA was then injected into the thorax of two- to four-day-old cold-anesthetized female mosquitoes. 69 nl of 3 μg / μl dsRNA was used for functional evaluation of the 14 3'UTR-interacting proteins in whole mosquitoes and 138 nl of 14 μg/μl of AeStaufen dsRNA was used for functional evaluation of AeStaufen in different tissues, including the salivary glands. We injected a higher quantity of dsRNA against AeStaufen to have an effective silencing in salivary glands. The same amounts of dsLacZ were injected as control in both experiments.

## Gene expression quantification

Total RNA was extracted from 10 mosquitoes or 10 organs using the E.Z.N.A. Total RNA kit I, DNase-treated using the TURBO DNA-free kit (Thermo Fisher Scientific) and reverse-transcribed using the iScript cDNA Synthesis Kit (Bio-Rad). Gene expression was quantified using the SensiFAST SYBR No-ROX Kit (Bioline) with the primers listed in S4 Table. The mRNA levels of *Actin*, a house-keeping gene, was quantified to normalize mRNA amount. Quantification was conducted on a CFX96 Touch Real-Time PCR Detection System with the following thermal profile: 95˚C for 1 min and 40 cycles of 95˚C for 10 sec and 60˚C for 15 sec, followed by melting curve analysis. Three repeats were conducted.

## Quantification of focus forming units

Single whole mosquitoes were homogenized in 500 μl of RPMI medium with silica beads (BioSpec) using mini-beadbeater (BioSpec). Homogenized tissues and saliva were sterilized by passing them through 0.22 μm filter (Sartorius). Samples were subjected to a 10-fold serial dilution in RPMI medium and 150 μl of each dilution was incubated with $1.5 \times 10^5$ C6/36 cells for 1 h, with gentle rocking at every 15 min. The inoculum was then removed and 1 ml of 1%

carboxymethyl cellulose (CMC) (Aquacide II, Calbiochem) and 2% FBS diluted in RPMI medium was added. After 3 days, the CMC medium was removed and the cells were fixed with 4% paraformaldehyde (Merck), permeabilized with 0.5% Triton X-100 (Sigma), blocked with 2% FBS, and stained using 1:400 mouse monoclonal anti-envelope antibody (4G2) and 1:20,000 secondary anti-mouse Dylight 680 (Rockland). Focus forming units (ffu) were counted using the Odyssey Clx imaging system (LI-COR) in three replicates per dilution and the average ffu per sample was calculated. Infection rate corresponded to the number of samples with at least one ffu over the total number of tested samples.

## siRNA-mediated gene silencing

The siRNAs targeting HsPURA [SI00696066 (A1), SI04175367 (A2) and SI04342744 (A3)] and siRNAs targeting HsPURB [SI04176879 (B1), SI04357661 (B2), SI04376281(B9)] were obtained from Qiagen. The siRNA negative control (siNC) ON-TARGETplus siRNA#2 was obtained from Dharmacon (Horizon Discovery). A final concentration of 5 nM siRNA was complexed with 2.5 μl of Lipofectamine RNAiMax (Invitrogen) and incubated in 12-well plates for 15 minutes prior to the plating of $1.3 \times 10^5$ Huh7 cells. Forty-eight hours post-transfection, cells were infected with DENV NGC at a MOI of 0.1. At 24 hours post-infection (hpi) plaque assay was performed on the supernatant as previously detailed [27], and at 48 hpi, western blot analysis was conducted on cell lysates.

## CRISPR/Cas9-mediated knockout

Two short-guide RNAs (sgRNA) targeting the sequences 5'-CGAG-CAGGGTGGTGCGGCGC-3' and 5-CGGCGGCGAGCAAGAGACGC-3' corresponding to *HsPURA* and *HsPURB*, respectively, were designed with an online CRISPR tool (crispr.mit.edu). The sgRNAs were cloned into pSpCas9-BB-2A-GFP (Adgene) as previously detailed [63]. $2.3 \times 10^5$ Huh-7 cells were transfected with 2 μg of pSpCas9-BB-2A-GFP-sgPURA using Lipofectamine 2000 (Invitrogen) following the manufacturer's instructions. At 24 h post-transfection, cells expressing GFP were sorted on a BD FACSAria II flow cytometer (BD Biosciences) and single cells were isolated by serial dilution to generate clonal populations of HsPURA knockout (KO) cells. HsPurA depletion was validated using western blotting. One of the PURA KO clones was then transfected with pSpCas9-BB-2A-GFP-sgPURB following the same procedure as for generating HsPurA KO. Four double HsPURA/HsPURB KO clones were produced. PURA/B KO cells were infected as detailed for siRNA-silenced cells.

## Intrathoracic inoculation

Four days post-injection of dsRNA against *AeStaufen*, 0.035 pfu of DENV2 were inoculated into the thorax of individual female mosquitoes using Nanoject II (Drummond scientific company). Mosquitoes were analyzed 7 days post-inoculation, assuming a similar incubation period as that of 10 days post-oral infection, as the midgut stage is bypassed during intrathoracic inoculation.

## Saliva collection

At seven days post-inoculation or ten days post-oral infection, mosquitoes were immobilized by cutting their wings and legs and their proboscises were inserted individually into 20 μl tips, containing 10 μl of equal volume of RPMI medium and SPF pig blood. After 30 min, mosquitoes were visually observed for the presence of blood in their abdomen. To prevent false negatives, we analyzed saliva only from mosquitoes in which blood could be visually detected,

indicating feeding and, hence, salivation. Salivation rate was calculated by dividing the number of salivating mosquitoes over the number of living mosquitoes.

## Statistics

Percentages for infection, blood-feeding and survival rates were calculated by pooling data from several experiments and by calculating the overall percentages based on the actual mosquito numbers. Standard error for percentages were estimated by calculating the square-root of [p (1-p)/n], where p is the proportion and n the number of observations. Differences in percentages were evaluated with the comparative error test. To justify pooling results from different experiments and ruling out any uncontrolled variable, we validated that controls were not significantly different. Z-test, unpaired T-test, and Dunnett's test were conducted with Prism 8.0.2 (GraphPad). Copies of gRNA and titers (FFU) were log-transformed before statistical analysis to meet normal distribution.

## Supporting information

**S1 Fig. Validation of anti-human PurB antibody to recognize AePur in *Aedes aegypti* mosquitoes.** Mosquitoes were injected with either dsRNA against AePur or a dsRNA control against LacZ. Eleven days post-injection, eight mosquitoes were homogenized in RIPA and protein quantity was quantified with microBCA. The same quantity of protein was used for WB and revealed with anti-human PurB. Analysis was done in triplicate and each column represents a replicate.
(TIFF)

**S2 Fig. AeStaufen-V5 (AeStau-V5) overexpression in C6/36 mosquito cells. (A)** Illustration of the AeStaufen-pIZT/V5 plasmid construct. **(B)** Western Blot of AeStaufen-V5 (AeStau-V5) at 24, 48 and 72h post-transfection in C6/36 cells. Chloramphenicol Acetyltransferase (CAT)-V5 provided in the kit used to control for expression. Actin was used as loading control.
(TIFF)

**S3 Fig. Quantification of RNAi-mediated gene silencing in mosquitoes.** Adult female *Ae. aegypti* mosquitoes were intrathoracically injected with dsRNA targeting the indicated genes or a dsRNA control targeting *LacZ* gene. Gene expression was quantified by RT-qPCR four days later in pools of five whole mosquitoes. Data show mean ± s.e.m. from three independent repeats. Reduction in percentage is indicated on the column for the targeted genes. *, p-value < 0.05; ***, p-value < 0.001, as determined by unpaired T-test.
(TIFF)

**S4 Fig. Impact of gene silencing on blood feeding and the survival of infected mosquitoes.** Adult female *A. aegypti* mosquitoes were intrathoracically injected with dsRNA targeting the indicated genes or a dsRNA control targeting the LacZ gene. Four days post-dsRNA injection, the mosquitoes were exposed to a blood meal containing $10^6$ pfu/ml of DENV. **(A)** Blood feeding rate. **(B)** Survival rate determined at 7 days post-feeding. Bars indicate percentage ± s.e. N, number of mosquitoes used for oral feeding in several biological repeats.
(TIFF)

**S5 Fig. Clustal W amino acid alignment of AePur, HsPURA, HsPURB, HsPurG-1 and HsPurG-2.** The three repeats of PUR domain are indicated. Conserved amino acids are highlighted. Color coding as follows: yellow, aromatic (F,W, Y); red, acidic (D, E); blue, basic (R, H, K), orange, non-polar (A, G, I, L, M, P, V); and green, polar (C, N, W, S, T).
(TIF)

**S6 Fig. Validation of *AeStaufen* silencing.** AeStaufen gene expression was quantified at 10 days post-oral infection in pools of ten whole mosquitoes, carcasses, midguts or salivary glands. *Actin* expression levels were used for normalization. Bars show mean ± s.e.m. from three repeats.
(TIF)

**S7 Fig. Homology of Staufen proteins in *Ae. aegypti*, *D. melanogaster* and *Homo sapiens*.** **(A)** Positions of the RNA-binding domains (RBD) in Staufen homologs. TBD, Tubulin-binding domain. SSM, Staufen-swapping motif. **(B-F)** Alignment of RBD1-5 in different species.
(TIFF)

**S1 Table. Details of all the proteins detected by RNA-affinity chromatography.** Uniprot IDs, fold changes and p-values are systematically provided.
(XLS)

**S2 Table. Effect of *AeStaufen* silencing on blood feeding, survival, and salivation rates.**
(DOCX)

**S3 Table Primer pairs used for dsRNA production.**
(DOCX)

**S4 Table. Primers used for Real-Time qPCR.**
(DOCX)

## Acknowledgments

We are very thankful to Mariano Garcia-Blanco from University of Texas Medical Branch (UTMB) for his support. We thank members of the Garcia-Blanco-Pompon Laboratory (Duke-NUS) for their important suggestions and support. We also thank the scientific editor Ravinuthula Sruthi Jagannathan for editing. We also thank the Mass Spectrometry Facility at UTMB for conducting the MS analyses.

## Author Contributions

**Conceptualization:** Julien Pompon.

**Formal analysis:** Shih-Chia Yeh, Mayra Diosa-Toro, Julien Pompon.

**Funding acquisition:** Julien Pompon.

**Investigation:** Shih-Chia Yeh, Mayra Diosa-Toro, Wei-Lian Tan, Florian Rachenne, Arthur Hain, Celestia Pei Xuan Yeo, Inès Bribes, Benjamin Wong Wei Xiang, Gayathiri Sathia-moorthy Kannan, Menchie Casayuran Manuel.

**Methodology:** Shih-Chia Yeh, Mayra Diosa-Toro, Dorothée Missé, Yu Keung Mok, Julien Pompon.

**Project administration:** Julien Pompon.

**Resources:** Dorothée Missé.

**Supervision:** Yu Keung Mok, Julien Pompon.

**Visualization:** Shih-Chia Yeh, Mayra Diosa-Toro, Julien Pompon.

**Writing – original draft:** Shih-Chia Yeh, Mayra Diosa-Toro, Julien Pompon.

**Writing – review & editing:** Shih-Chia Yeh, Mayra Diosa-Toro, Dorothée Missé, Yu Keung Mok, Julien Pompon.

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
