## [Decision Letter · Decision Letter 0]

19 Apr 2022

Dear Dr Pompon,

Thank you very much for submitting your manuscript "Characterization of dengue virus 3’UTR RNA binding proteins in mosquitoes reveals that AeStaufen reduces subgenomic flaviviral RNA in saliva" for consideration at PLOS Pathogens. As with all papers reviewed by the journal, your manuscript was reviewed by members of the editorial board and by several independent reviewers. The reviewers appreciated the attention to an important topic. Based on the reviews, we are likely to accept this manuscript for publication, providing that you modify the manuscript according to the review recommendations.

While I have considered this a 'minor' revision - one of the reviewers listed a number of major concerns that could be rectified with additional data/analysis. You will need to address at least some of these concerns.

Sincerely,

Elizabeth A. McGraw, PhD

Associate Editor

PLOS Pathogens

Sonja Best

Section Editor

PLOS Pathogens

Kasturi Haldar

Editor-in-Chief

PLOS Pathogens

orcid.org/0000-0001-5065-158X

Michael Malim

Editor-in-Chief

PLOS Pathogens

orcid.org/0000-0002-7699-2064

While I have considered this a 'minor' revision - one of the reviewers listed a number of major concerns that could be rectified with additional data/analysis. You will need to address at least some of these concerns.

Reviewer Comments (if any, and for reference):

Reviewer's Responses to Questions

**Part I - Summary**

Reviewer #1: The host factors that positively or negatively modulate dengue virus infection in mosquitoes are still poorly characterized. The authors used RNA-affinity chromatography and mass spectrometry to identify RNA-binding proteins interacting with the dengue virus 3’UTR in Aedes aegypti cells, followed by their functional validation in mosquitoes in vivo. This is a well conducted study, and the manuscript is clearly organized and written. The topic is relevant and the approach is innovative. My (mostly minor) suggestions to improve the paper are below.

Reviewer #2: The genome and the replicative forms of Dengue viruses, as is the case with other RNA viruses, is able to interact with RNA binding proteins (RBPs). While the importance of RBPs has been documented in humans, there is poor information about the relevance of these factors in DENV transmission by mosquitoes. In the present manuscript authors identified RBPs which interact with DENV2 3’UTR in Ae. aegypti cells. Authors confirmed the interaction of AePur to the 3’UTR, whereas AeStaufen interacts with both 3’UTR and sfRNA. The proviral function of AeRan, AeExoRNase, and AeRNase was evaluated by in vivo functional assay, whereas AeGTPase, AeAtu, and AePur have anti-viral functions in mosquitoes.

Authors also found that AeStaufen mediates a reduction of gRNA and

sfRNA copies in several mosquito tissues, revealing AeStaufen’s role in DENV transmission.

The manuscript is interesting and relevant, however there are some aspects that should considered to fully support authors conclusions.

Reviewer #3: “Characterization of dengue virus 3’UTR RNA binding proteins in mosquitoes reveals that AeStaufen reduces subgenomic flaviviral RNA in saliva” by Shih-Chia Yeh et al.,

The main findings: DENV shuttle between vertebrate hosts and mosquitoes. Although many human RNA-binding proteins (RBPs) have been identified with important roles during DENV life cycle in human cells, the role of mosquito RBPs in regulating DENV infection cycle is less investigated. Using RNA-affinity chromatography followed by mass spectrometry, the authors report in this paper on the identification of fourteen Aedes aegypti proteins which associate with DENV2 3’UTR. They then choose two candidates to verify these associations. They further conduct in vivo functional analysis of these candidates in mosquito and identify 3 pro-viral candidates (AeRan, AeExoRNase and AeRNase) and 3 anti-viral candidates (AeGTPase, AeAtu and AePur). Interestingly, though human homologs of AePur also associate with DENV2, these human proteins do not have a role in DENV2 infection in the cultured cells. Whereas the initial investigation by RNAi in mosquito does not identify a role of AeStaufen in DENV2 infection, detailed analyses of AeStaufen function in mosquito tissues show that it suppresses both gRNA and sfRNA levels in several tissues including the salivary gland. Of interest, AeStaufen reduces sfRNA:gRNA ratio in the saliva without affecting viral particles.

Novelty/significance: As a single-strained RNA virus, the life cycle of DENV in mosquito is tightly regulated by mosquito factors. Among these factors, RNA binding proteins play important roles in controlling DENV gRNA translation, degradation, replication, and virion assembly. Although similar approach has already being developed to identify Aedes proteins which bind to 3’UTR of Zika virus, this study is the first attempt to identify Aedes proteins associate with DENV2 3’UTR and provides some useful information for follow up study. Overall, the study is well designed and this is an interesting piece of work with novel aspects. It offers some insights on how DENV2 infection and replication in mosquito is regulated via its 3’UTR associated proteins, which warrant further investigation.

Weakness: Some methodology is not well explained. Some texts are a bit confusing and need to be rephrased. Additional data could be considered to support the authors’ claim on the “transmission inhibition” role of AeStaufen. Addressing/clarifying these would strengthen the conclusion present in this manuscript.

**Part II – Major Issues: Key Experiments Required for Acceptance**

Reviewer #1: (No Response)

Reviewer #2: Major concerns

1. Although the RNA-affinity chromatography coupled with mass spectrometry (MS) using mosquito cell lysates is an appropriate strategy to analyze RBPs, authors only used uninfected mosquito cell lysates. It is possible that the RBPs identified from uninfected cell lysates may differ from the ones identified in infected cell lysates. The validation of the RBPs should be performed in the presence of infected cell lysates.

2. How do the authors know that the RBPs identified are bound directly to the RNA? Is it possible that the proteins are not binding directly to the RNA and are part of a complex which is bound to the RNA?

3. Have the authors confirmed the interaction of the identified proteins with gRNA of sfRNA by mobility shift assays?

4. Have the authors analyzed the location of the identified RBPs in infected cells? Have these proteins the same location than viral RNA?

5. Have the authors confirmed some of the interactions using the 3’UTR of another DENV serotype? Have the authors analyzed the role of AeStaufe in the infection with others DENV serotypes in mosquito?

Reviewer #3: In several places (line 40-41 in Abstract, line 62-65 in Author abstract, line 337-338 and line 431-432 in Discussion), the authors state that “AeStaufen impacts on viral transmission”. However, data present in this manuscript are in line with their claim that “AeStaufen reduces subgenomic flaviviral RNA in saliva” (in the title) but do not lend support on its role on viral transmission. Interestingly, saliva from AeStaufen knockdown mosquito show higher sfRNA:gRNA ratio without affecting viral titers. This provides a unique situation to test whether AeStaufen has an impact on viral transmission (via controlling the sfRNA:gRNA ratio). The authors could use the infection assay established in their ref 28 to infect human cells with saliva collected from control or AeStaufen knockdown mosquitoes. If saliva derived from AeStaufen knockdown is more infectious than saliva from control knockdown, it will not only demonstrate the role of AeStaufen during DENV2 infection cycle but also give strong support to the role of sfRNA during virus infection in general.

**Part III – Minor Issues: Editorial and Data Presentation Modifications**

Reviewer #1: Main comments:

In the RNA immunoprecipitation assays (Figure 2), the sfRNA/gRNA ratio in immunoprecipitates could simply reflect their ratio in the protein lysate, not the differential binding affinity of candidate RBPs. This analysis should account for the original ratio, which is presumably different between live mosquitoes and C6/36 cells.

The authors claim that they “validated gene silencing” (line 195) but the data presented in Figure S2 indicate that silencing efficiency was highly variable among target genes. It would be useful to report the % reduction in gene expression level and its statistical significance, as well as discuss the implications of incomplete silencing for some of the candidate genes.

There are some statistical issues with the analysis of proportions (infection rate, feeding rate, survival rate) such as those shown in Figure 3A and Figure S3. In Figure 3A, the use of Z test does not seem appropriate to compare infection rates because Z test is only valid for a continuous variable with a large sample size and a normally distributed residual variance. By nature, percentages cannot be normally distributed because they have lower and upper limits (0 and 100). Moreover, calculating the average of percentages is incorrect unless the three experimental repeats had the exact same sample size, or if the average was weighted. In general percentages are more safely analyzed as a binary variable with a 95% confidence interval, which directly accounts for the sample size.

Pooling experimental repeats (for example, Figure 3, Figure S3) could be misleading if the repeats were inconsistent. In general, pooling is only justified when a potential experiment effect (or experiment x treatment interaction effect) has been ruled out.

The authors could have discussed to which extent the choice of DENV type and strain (DENV2 NGC) could have influenced the results. All four DENV types share conserved functional RNA structures in their 3’UTR (except for DENV4), however the 3’UTR differs in sequence length and nucleotide composition between DENV types.

Minor comments:

Line 106: “interphase”, do you mean “interface”?

Line 210: Expressing a proportion as percentage of reduction is ambiguous because it could represent the absolute difference in percentage or the percentage of reduction, relative to the control. Please clarify.

Lines 249-250: Since the authors did not test HsPurG this statement could be softened.

Line 396: “non-mediated”, do you mean “nonsense-mediated”?

Line 506: Is the blood meal titer stable for 1.5 hours?

Reviewer #2: None

Reviewer #3: 1. Among the fourteen proteins identified, some of them (such as AeGTPase) do not contain defined/known RNA binding domain. These proteins may associate with DENV2 3’UTR indirectly via other proteins and do not necessarily have a direct interaction with DENV2 3’UTR. Thus, it is more appropriate to use “association” than “binding” to describe the relation between these identified proteins and DENV2 3’UTR.

2. Following sentences are a bit confusing and misleading. The authors might like to rephrase them.

• Line 141-142 “AeMaleless, an ATP-dependent helicase involved in dosage compensation [30]”. This sentence gives the impression that AeMaleless plays a role in dosage compensation. In fact, the role of AeMaleless has not been established. The role of Maleless in dosage compensation is reported in Drosophila melanogaster.

• Similarly, in Line 142-142 “AeSex-lethal, an RNA-binding protein also involved in dosage compensation” need to be rephrased.

3. For each of fourteen candidate genes, ~0.2 ug of dsRNA was injected into mosquito to evaluate their potential function, while ~1.9 ug of dsRNA was delivered to address AeStaufen function in different tissues. The authors should explain the reason of using different amount of dsRNA in these experiments.

4. Line 201-202, “we controlled that survival was similar across the various silencing conditions”. How did the authors control the survival rate without affecting the knockdown efficiency? Similarly, in Line 270, “we controlled that blood feeding and survival rates were not affected by AeStaufen depletion…”. How was this experimented conducted without affecting knockdown efficiency? Line 298 “we controlled that the salivation rate was not affected by AeStaufen silencing…”. How was salivation rate experimentally controlled without affecting knockdown efficiency? The authors need to add the methodology used for these experiments.

5. For the RIP experiment, anti-human PurB antibody is used to pull down AePur, it’d be great if the authors could conduct dsRNA knockdown experiment in cells to verify the specificity of this antibody toward AePur.

6. There is a discrepancy of DENV2 titer used for AePur RIP experiment. It is mentioned as 10^6 PFU/mL for AePer RIP (line 502-505), but it is stated as 10^7 PFU/mL. The authors need to correct one mistake here.

PLOS authors have the option to publish the peer review history of their article (what does this mean?). If published, this will include your full peer review and any attached files.

Reviewer #1: **Yes: **Louis Lambrechts

Reviewer #2: No

Reviewer #3: No

Figure Files:

Data Requirements:

Reproducibility:

References:

---

## [Editor Report · Decision Letter 1]

9 Sep 2022

Dear Dr Pompon,

We are pleased to inform you that your manuscript 'Characterization of dengue virus 3’UTR RNA binding proteins in mosquitoes reveals that AeStaufen reduces subgenomic flaviviral RNA in saliva' has been provisionally accepted for publication in PLOS Pathogens.

Best regards,

Elizabeth A. McGraw, PhD

Associate Editor

PLOS Pathogens

Sonja Best

Section Editor

PLOS Pathogens

Kasturi Haldar

Editor-in-Chief

PLOS Pathogens

orcid.org/0000-0001-5065-158X

Michael Malim

Editor-in-Chief

PLOS Pathogens

orcid.org/0000-0002-7699-2064

While most of the suggestions do not rise to the level of a major revision, there are many things raised by the reviewers. Please attempt to address them all.
---

## [Editor Report · Acceptance letter]

14 Sep 2022

Dear Dr Pompon,

We are delighted to inform you that your manuscript, "Characterization of dengue virus 3’UTR RNA binding proteins in mosquitoes reveals that AeStaufen reduces subgenomic flaviviral RNA in saliva," has been formally accepted for publication in PLOS Pathogens.

Best regards,

Kasturi Haldar

Editor-in-Chief

PLOS Pathogens

orcid.org/0000-0001-5065-158X

Michael Malim

Editor-in-Chief

PLOS Pathogens

orcid.org/0000-0002-7699-2064